# Study of Membrane-Immobilized Oxidoreductases in Wastewater Treatment for Micropollutants Removal

**DOI:** 10.3390/ijms232214086

**Published:** 2022-11-15

**Authors:** Agata Zdarta, Jakub Zdarta

**Affiliations:** 1Institute of Chemical Technology and Engineering, Faculty of Chemical Technology, Poznan University of Technology, Berdychowo 4, PL-60695 Poznan, Poland; 2Circularity & Environmental Impact, Department of Environmental and Resource Engineering, Technical University of Denmark, Bygningstorvet 115, 2800 Kgs. Lyngby, Denmark; 3Process and Systems Engineering Centre, Department of Chemical and Biochemical Engineering, Technical University of Denmark, Building 229, 2800 Kgs. Lyngby, Denmark

**Keywords:** enzyme immobilization, oxidoreductase, membranes, enzymatic membrane reactors, wastewater treatment, bioremoval

## Abstract

The development of efficient strategies for wastewater treatment to remove micropollutants is of the highest importance. Hence, in this study, we presented a rapid approach to the production of biocatalytic membranes based on commercially available cellulose membrane and oxidoreductase enzymes including laccase, tyrosinase, and horseradish peroxidase. Effective enzyme deposition was confirmed based on Fourier transform infrared spectra, whereas results of spectrophotometric measurements showed that immobilization yield for all proposed systems exceeded 80% followed by over 80% activity recovery, with the highest values (over 90%) noticed for the membrane-laccase system. Further, storage stability and reusability of the immobilized enzyme were improved, reaching over 75% after, respectively, 20 days of storage, and 10 repeated biocatalytic cycles. The key stage of the study concerned the use of produced membranes for the removal of hematoporphyrin, (2,4-dichlorophenoxy)acetic acid (2,4-D), 17α-ethynylestradiol, tetracycline, tert-amyl alcohol (anesthetic drug), and ketoprofen methyl ester from real wastewater sampling at various places in the wastewater treatment plant. Although produced membranes showed mixed removal rates, all of the analyzed compounds were at least partially removed from the wastewater. Obtained data clearly showed, however, that composition of the wastewater matrix, type of pollutants as well as type of enzyme strongly affect the efficiency of enzymatic treatment of wastewater.

## 1. Introduction 

Substances acting as environmental pollutants are continuously and uncontrollably released into the environment possessing a threat to organisms living in it. A particular source of contamination is wastewater, carrying pollution from homes, household chemicals, personal care products, pharmaceuticals, organic pollutants, and other potentially toxic substances [1]. Their removal is especially challenging due to their variety of sizes, chemical structures, and properties. Among wastewater treatment methods, membrane processes gather more and more interest as effective ways to pretreat or remove contaminants from the wastewater. The main advantages of membrane processes are low capital cost, minimalization of the energy requirement, and reduction of the equipment size [2]. This gives the membrane technologies application potential as an economical and sustainable tool for wastewater treatment. However, they act rather as pollutants separators, failing in the degradation of the contaminants [3]. A promising method for selective elimination of emerging contaminants provides enzyme technology. Enzymes can provide a complete decomposition of contaminants, e.g., through an oxidation mechanism (oxidative enzymes), but their short life and sensitivity raise even more initially the high cost of their use [4].

The solution to these problems may be to combine the aforementioned techniques, creating enzymatic membrane bioreactors for efficient separation and degradation of contaminants. Such systems based on oxidoreductases have been tested for dyes [5], chemicals [6], and pharmaceuticals removal [7]. For example, cross-linked laccase aggregates obtained on ZnO/SiO_2_ composite support were achieved in a study conducted by Sun et al. [5], with more than 80% removal efficiency of Acid Black 10BX and Acid Blue 113 dyes. The same enzyme derived from *T. versicolor* was immobilized on *H. communis* sponge scaffolds and used to biodegrade bisphenols. Under optimal conditions, i.e., pH 5 and temperature, 30 °C and 40 °C, respectively, bisphenol A (BPA) and bisphenol F (BPF) were completely removed, and more than 40% of resistant to biodegradation bisphenol S (BPS) was degraded. The immobilized laccase further showed good reusability even after five biocatalytic cycles of bisphenol removal [6]. Furthermore, Nguyen et al. [7] reported more than 90% removal efficiency from wastewater of diclofenac, sulfamethoxazole, and carbamazepine using laccase immobilized on granular activated carbon. The results published are promising, although the biggest challenge is the selection of an immobilization matrix, providing enzyme protection, while at the same time inhibiting its catalytic capacity as little as possible [2]. Moreover, research to date focused mainly on the removal of single compounds or groups of similar compounds from the model solution. Additionally, the application of membrane enzymatic reactors for pollutant removal is not a widely studied topic.

As the literature lacks papers describing enzymatic membrane reactors (EMRs) application in more complex solutions originating from real wastewater, the present study aimed to produce biocatalytic membranes based on immobilized laccase, tyrosinase, and horseradish peroxidase, and to determine the efficiency of the enzymatic wastewater treatment process in a membrane reactor in the removal of hematoporphyrin, (2,4-dichlorophenoxy)acetic acid, 17α-ethynylestradiol, tetracycline, tert-amyl alcohol, and ketoprofen methyl ester. In the study, wastewater before purification in a wastewater treatment plant (wastewater ‘before’), and after that process (wastewater ‘after’) was tested in order to determine how the sampling place affects the removal of analyzed micropollutants.

## 2. Results

### 2.1. Enzyme Immobilization

The first step of obtaining biocatalytic membranes for application in EMRs is efficient enzyme immobilization. Thus, in the first step of the investigation, Fourier transform infrared spectroscopy (FTIR) was used to confirm enzyme deposition. In Figure 1, FTIR spectra of the produced systems, as well as spectra of pristine cellulose membrane and used enzyme, for comparison, are presented. The FTIR spectra of the pristine membrane made of regenerated cellulose showed the presence of the wide absorption peak with a maximum around 3490 cm^−1^ assigned to stretching vibrations of -OH groups, signals around 2950 cm^−1^ and 2850 cm^−1^ characteristic for C-H stretching vibrations, band around 1750 cm^−1^ assigned to C=O stretching vibration and a peak with a maximum around 1050 cm^−1^ from stretching vibrations of C-O bonds. The presented FTIR spectra of all of the analyzed enzymes showed the presence of very similar signals with slight shifts in the wavenumber value of their maxima. These signals are characteristic of the peptide structure of the enzyme. Among the most important, signals around 3420 cm^−1^ (stretching -OH), peaks at 1655 cm^−1^ and 1545 cm^−1^ assigned to stretching vibrations of amide I, and amide II bands, respectively, as well as bands with maxima around 1150 cm^−1^ and 1050 cm^−1^ (stretching C-O-C and C-O in the enzyme carbon chain), can be seen. FTIR spectra obtained for the membranes after enzyme deposition exhibited signals characteristic for both cellulose support and deposited enzyme, indicating efficient enzyme immobilization. Further, slight shifts of the signal’s maxima in spectra upon immobilization, as compared to free enzyme indicate changes in the enzyme microenvironment upon biocatalyst binding.

To deeper investigate the immobilization process, its yield, membrane loading, and amount of deposited enzyme were determined (Table 1). It can be seen that immobilization yield, calculated from the activity of the free enzyme and supernatant after immobilization reached over 80% for performed processes. The highest yield (91%) was observed for the membrane with immobilized laccase, whereas the lowest (82%) was noticed for the membrane with tyrosinase. Consequently, the amount of immobilized enzyme followed the same trend. 9.1 mg of the laccase was immobilized onto/into the membrane, while for HRP and tyrosinase this value reached 8.5 mg and 8.2 mg, respectively. This amount of immobilized enzyme corresponds to relatively low membrane loading which reached 0.68 mg/cm^2^ for membrane-laccase system, 0.63 mg/cm^2^ for membrane-HRP, and 0.61 mg/cm^2^ system with immobilized tyrosinase. Hence, although immobilization yields differ from each other by around 5–10%, the membrane loading is very similar in each of the produced systems.

### 2.2. Characterization of Biocatalytic Membranes Produced

After confirmation of efficient immobilization, activity recovery of the produced systems was examined as a key feature of the immobilized enzymes (Table 2). It can be seen that all produced membranes showed over 80% activity recovery. The highest activity was characterized as a system with immobilized laccase (90%), whereas the lowest was a membrane with deposited horseradish peroxidase (82%). Nevertheless, catalytic systems characterized by over 80% activity recovery should be considered systems capable of application in EMRs.

During the course of the study, the kinetic parameters of free and immobilized enzymes were also determined in order to follow changes in catalytic performance before and upon immobilization (Table 2). All analyzed enzymes upon immobilization showed an increase in Michaelis-Menten constant values and a simultaneous drop in maximum reaction rate. The most prominent changes in *K_M_* value were observed for the HRP enzyme, which *K_M_* raised from 1.33 mM (free enzyme) to 1.52 mM (immobilized enzyme), whereas the lowest changes in *K_M_* were noticed for the laccase enzyme (an increase from 0.087 mM for free laccase to 0.094 mM for membrane-bounded laccase). Similar changes were observed for *V_max_* data. The most significant drop was observed for the HRP enzyme and reached about 20%, from 374 mM/min for free HRP to 298 mM/min for immobilized one. On the other hand, the lowest decrease in maximum reaction rate was noticed for the laccase enzyme. The *V_max_* of this enzyme upon immobilization reached 0.103 mM/min and was less than 10% lower than the *V_max_* of the free enzyme (0.112 mM/min). Presented results clearly indicate that immobilized enzymes showed lower substrate affinity (higher *K_M_* values) which resulted in a lower maximum reaction rate, as compared to free enzymes.

In the production of efficient biocatalytic membrane-based systems for application in EMRs, it is also crucial to determine permeability (*L_p_*) and flux (*J*) of the materials, as these parameters affect process duration and consequently its costs. Pristine ultrafiltration Ultracel 5 kDa membrane made of regenerated cellulose is characterized by the water permeability of 12.17 L/m^2^·bar·h and flux of 48.69 L/m^2^·h (measured at 4 bar) that stays in agreement with previously published data [8]. From the obtained data, it is clear that all membranes after enzyme immobilization showed a drop in water permeability and flux (Figure 2), clearly suggesting enzyme immobilization onto membrane surface and into membrane pores [9]. For instance, an almost 50% drop in water permeability and flux, to 6.57 L/m^2^·bar·h and 26.29 L/m^2^·h, respectively, was noticed for the membrane after laccase immobilization showing that enzyme immobilization might significantly affect the properties of the membrane. By contrast, water permeability of the membrane-HRP system decreased less than 20%, to 39.43 L/m^2^·bar·h.

Furthermore, we have also followed changes in water permeability and flux of all produced membranes upon wastewater treatment process as components of wastewater matrix might lead to a drop in these parameters. Indeed, all membranes after wastewater treatment showed lower values of both analyzed parameters. Membranes used for the treatment of ‘before’ wastewater exhibited around 40% lower values of *L_p_* and flux, as compared to the membranes upon immobilization. The lowest value of water permeability and flux was characterized by a system with immobilized tyrosinase, showing *L_p_* of 5.29 L/m^2^·bar·h and *J* of around 21.2 L/m^2^·h. However, it should be underlined that membranes with around 10% higher water permeability and flux were characterized after treatment of wastewater ‘after’, as compared to the wastewater ‘before’.

Biocatalytic systems suitable for practical and large-scale applications should also be characterized by long-term storage stability as well as great reusability, in order to minimize process costs related to enzyme use. In the study, the activity of the enzymes immobilized using membranes over storage time, and repeated use was determined using a model oxidation reaction of ABTS. The storage stability of the produced membranes was examined over 20 days, whereas their recycle potential, followed by changes in enzyme loading, was determined over 10 cycles (Figure 3). From Figure 3a, it is clear that long-term stability of all produced membranes declined with prolongation of storage time. After 20 days of the test, membrane-laccase, membrane-tyrosinase, and membrane-HRP retained, respectively, 83%, 80%, and 74% of the relative activity. It was over 40% higher compared to the activity of free enzymes after 20 days of storage (data not presented). Hence, this data clearly showed that the long-term stability of the biocatalysts was improved upon immobilization using membranes.

However, the biggest advantage of the immobilized enzymes is the possibility of their multiple uses without complicated separation and purification procedures. The relative activity of the immobilized biocatalysts dropped gradually over repeated use, however, all produced membrane-enzyme systems exhibited great reusability, as even after 10 reuse cycles showed over 65% of their initial activity (Figure 3b). By the highest relative activity (78%), retention was characterized immobilized laccase, while the activity of the immobilized tyrosinase and HRP were slightly lower and after repeated use reached 72% and 64%, respectively. The values of activity retention over reuse cycles are closely related to data on enzyme loading, as enzyme leaching from the support and its inactivation should be considered the main factors affecting enzyme multiple uses. In the study, the amount of enzyme leaked from the support was determined according to Bradford methodology by spectroscopic analysis of permeates after each degradation cycle. It can be seen that only around 10% of the laccase was eluted from the cellulose membrane, whereas for tyrosinase and HRP, the leaching rate was higher and reached 18% and 20%, respectively. This shows that the type of the enzyme and formed enzyme-support interactions have a strong effect on enzyme binding and, consequently, also on activity retention over repeated use.

### 2.3. Removal of Micropollutants from Wastewater

Although previous data proved efficient production of catalytically active and stable membranes, their practical tests in the removal of selected micropollutants in EMRs were the most important part of this study. To prove the high effectivity of the proposed, systems in wastewater treatment real sewage from wastewater treatment plant were tested. Further, as the composition of the sewage and concentration of micropollutant vary depending on the treatment stage, sewage sampled at the entrance to the treatment plant (‘before’) and sewage leaving the wastewater treatment plant after finished purification process (‘after’) were used in the tests. The detailed composition of the analyzed wastewaters is presented in Appendix A, whereas the estimated concentration of compounds subjected to detailed analysis was given in Table 3. Due to the complex structure and the presence of numerous of various compounds in the wastewater, in this study, we have followed the fate of selected compounds from various groups of substances. The selected compounds, included: hematoporphyrin, a representant of porphyrins; (2,4-dichlorophenoxy)acetic acid (2,4-D), commonly used as a herbicide; 17α-ethynylestradiol (EE2), frequently used estrogen; tetracycline, used as an antibiotic; tert-amyl alcohol (anesthetic drug); and ketoprofen methyl ester, which occurs in nonsteroidal anti-inflammatory drugs. The selected compounds are usually known for their positive/therapeutic effect. However, long-term and uncontrolled contact with these substances might lead to undesired changes in ecosystems and living organisms [10,11]. Due to the fact that the above-mentioned compounds are considered micropollutants, there is a need to examine effective methods for their removal.

Prior to the examination of the micropollutants removal, treatment time was determined, as, in the real-scale application, process duration is one of the most important parameters determining the useability of applied biocatalysts. Figure 4 shows the duration of the wastewater treatment by produced systems tested in EMRs. It can be seen that, irrespective of the membraned used, the duration of treatment experiments using wastewater ‘before’ was about 10% longer, as compared to the treatment of ‘after’ wastewater, confirming previous observations that matrix of the wastewater sampled at the entrance to the treatment plant is more complicated. Their constituents might block membrane pores leading to lower membrane permeability and long process time. For instance, 26 min 33 s were required to treat wastewater ‘before’ by the membrane-tyrosinase system, whereas for wastewater ‘after’, this time was lower and reached 24 min 20 s. It should also be emphasized that less than 30 min were needed to perform the process supported by membrane-tyrosinase and membrane-HRP systems, whereas in the membrane-laccase system, process time was longer and exceeded 40 min, indicating that the shape and size of the enzyme molecule affect its deposition within the membrane. inconsequently, this also affects membrane permeability and process duration.

Afterward, in the presented study, we followed the removal of representants of various groups of micropollutants detected in tested sewage (Figure 5). From the presented data, it might be concluded that all tested biocatalytic systems showed mixed efficiency in the removal of examined micropollutants. Note, however, that all tested compounds were at least partially removed from the wastewater. Among tested membranes, the highest removal rates were observed for the membrane-laccase system. For this system, wastewater ‘after’ removal efficiency of 61% and 56% was obtained for 2,4-D and tetracycline, followed by 49% for EE2, 40% for ketoprofen methyl ester, 30% for tert-amyl alcohol, and 26% for hematoporphyrin. By contrast, up to 10% lower removal rates of micropollutants were noticed for wastewater ‘before’, clearly showing that sampling place and composition of wastewater matrix affect removal efficiency, while the lowest removal rates of all tested compounds were characterized in the membrane-HRP system. Nevertheless, the removal efficiency order did not change among membranes, showing that the type of degraded compound, its structure, and its resistance towards enzymatic conversion affect its removal rate.

Additionally, control experiments, with pristine membranes and membranes with thermally inactivated enzyme have been performed in order to determine the contribution of every single process in total removal efficiency. Collected data (Appendix A) clearly showed that rejection of pollutants by membrane ranged from 3% for hematoporphyrin to 10% for 2,4-D. Slightly higher results of pollutant removal were observed for membranes with the thermally inactivated enzyme. The highest removal by adsorption (17%) was observed for 2,4-D for the membrane-tyrosinase system, whereas the lowest (2%) for tert-amyl alcohol adsorbed on membrane-laccase and membrane-HRP system. Further, higher removal rates, of around 3% to 5% of all tested pollutants were noticed from the wastewater “after” treatment. Finally, from the obtained data, it is clear that catalytic conversion was the main driving mechanism of pollutants’ removal. The removal rate by membranes with active enzymes usually exceeded 20%, reaching up to 40% for the membrane-laccase system. By contrast, the lowest percentage contribution of catalytic removal was observed for membranes with immobilized HRP. In this case, the removal of hazardous compounds did not exceed 10%.

## 3. Discussion

### 3.1. Enzyme Immobilization and Characterization of Biocatalytic Systems Produced

The first stage of the presented study concerned the immobilization of three various enzymes, including laccase, tyrosinase, and HRP onto an ultrafiltration membrane with a skin layer made of regenerated cellulose. The application of membrane as enzyme support resulted several advantages, including enzyme deposition onto membrane surface and into its pores, as well as allowed direct application of the produced systems in enzymatic membrane bioreactors. In this way, simultaneous catalytic action and membrane separation could be performed resulting in higher process efficiency and improved purity of the final products [12]. However, to produce stable and active enzymatic systems, the support material is required to possess functional groups capable of enzyme binding. From FTIR spectra, it is clear that in the structure of cellulose hydroxyl and carbonyl groups occurred, whereas, in the structure of all of the analyzed enzymes, numerous amine groups from amino acids side chains are presented facilitating direct interactions between enzyme and support. In the FTIR spectra upon enzyme immobilization signals characteristic for used membrane and deposited biocatalyst can be seen, proving efficient immobilization, as reported earlier [13,14]. Nevertheless, slight shifts of the signals maxima in the spectra of produced systems, as compared to the pristine membrane and free enzyme, suggested minor changes in enzyme microenvironment and creation of mainly hydrogen bonds between enzyme and support as well as retention of catalytic activity by immobilized biocatalysts [15]. It should also be mentioned that a previous study concerning the use of membrane as enzyme support reported that enzyme might be partially deposited into membrane pores when membrane MWCO is properly adjusted to the size of the enzyme [16]. At least in this study, partial enzyme deposition into membrane pores cannot be excluded; however, a deeper investigation of this issue is not the main purpose of this research.

After confirmation of efficient enzyme deposition, it was important to follow changes in water permeability and flux of membranes before and after immobilization. Pristine Ultracel 5 kDa membrane is characterized by water permeability of 12.17 L/m^2^·h·bar, whereas for all produced systems, lower values of this parameter were noticed, suggesting enzyme deposition onto membrane surface and into membrane pores, as reported earlier [17,18]. The biggest drop in water permeability and flux was noticed in the membrane-laccase system, which might be related to the similarity in size of enzyme molecule and pores of the membrane that also facilitate enzyme immobilization into the pores [19]. The molecular size of the laccase from *Trametes versicolor* (around 50–55 kDa [20]) fits with the size of the membrane pores that allowed enzyme molecules to penetrate the membrane and deposit inside its pores [21]. Nevertheless, data collected on other produced membranes indicated also that size and shape of the enzyme, as well as the amount of immobilized enzyme and its deposition onto/into the membrane, stimulates permeability of the membrane and process duration.

To claim efficient immobilization, it is important to achieve high immobilization yield and high activity recovery. Immobilization yield of over 80% for each of the produced systems was obtained, with the highest reaching 91% for membrane-laccase. High immobilization yield was a result of numerous functional groups in membrane structure, as well as well-developed and organized porosity of the membrane. These features facilitate stable enzyme deposition and retention of high activity [2]. Additionally, the highest amount of immobilized enzyme (9.1 mg), and consequently the highest membrane loading (0.68 mg/cm^2^), were noticed for the membrane with immobilized laccase. As above-mentioned, this might be explained mainly by the laccase immobilization into membrane pores. Nevertheless, data obtained for tyrosinase and HRP enzymes should also be considered efficient. In both cases, over 8 mg of the enzyme were deposited within the membrane, which corresponds to membrane loading exceeding 0.6 mg/cm^2^. By contrast, in another study concerning the use of membranes for oxidoreductase immobilization, usually higher membrane loading was observed. Previously, over 5 mg/cm^2^ of enzyme loading was noticed when TiO_2_ blended the polyethersulfone membrane or nanofiltration membrane modified by polyelectrolytes via a layer-by-layer approach that was used as support for laccase immobilization [12,22]. However, high enzyme loading did not result in high activity recovery, whereas in both studies, less than 65% of activity recovery of the immobilized enzyme was observed. Proposed biocatalytic membranes showed over 80% activity recovery, making them promising systems with possible practical applications. The highest activity recovery (90%) was noticed for the membrane with immobilized laccase. This might be related to the fact, that among all enzymes examined, laccase is characterized by the highest mechanical resistance and tolerance toward changing conditions [23].

Comparison of changes in kinetic parameters of free and immobilized enzymes allowed us to determine changes in enzyme-substrate affinity upon immobilization and examine the maximum reaction rate. All produced systems showed around 10–15% higher values of *K_M_* constant, as compared to free counterparts, indicating lower substrate affinity towards immobilized biomolecules. This fact frequently occurred in enzyme immobilization and might be explained by several factors of both internal and external nature. Firstly, random immobilization might lead to the partial inaccessibility of the enzyme active sites [24]. Further, some unwanted conformational rearrangements might occur in the enzyme structure. Finally, mass transfer resistance might also lead to lower substrate affinity by the immobilized enzymes. Higher *K_M_* values resulted in around a 20% lower maximum reaction rate observed for the analyzed system. Besides lower substrate affinity, this fact might be explained by the formation of steric hindrance in the transfer of substrate and products, and the formation of diffusional limitations, as previously observed [25,26]. Temocin et al., reported even more prominent changes in values of kinetic parameters of HRP immobilized on electrospun poly (vinyl alcohol)–polyacrylamide nanofiber membrane. In this study, over 60% lower *V_max_* and over 30% higher *K_M_* were noticed [27]. In this study, the most significant changes in the values of kinetic parameters were also reported for the HRP enzyme indicating this enzyme as the most fragile towards changes occurring upon immobilization. Nevertheless, less than a 20% increase in the Michaelis-Menten constant and less than a 20% drop in maximum reaction rate caused the proposed systems to be considered highly active.

The biggest advantage of the immobilized enzyme is the possibility of their repeated use without significant loss of activity. This overcomes the costs of the support and immobilization procedure, promoting immobilized enzymes as an alternative for currently used catalysts. Although a progressive decline in relative activity over repeated use was observed after 10 cycles, tested systems retained from 78% (membrane-laccase) to 64% (membrane-HRP) of their initial properties. This data indicates improvement of biocatalyst stability over repeated use due to the protective effect of support material, including immobilization into membrane pores, protection against the negative effect of process conditions, and stabilization of enzyme structure [28,29]. The drop in relative activity might be explained by partial enzyme inactivation and/or its inhibition over repeated use as well as partial enzyme elution from the support, as suggested by the results of the elution study. Enzyme loading within the membrane followed the same trend as enzyme activity with the highest loading for membrane-laccase and the lowest for membrane-HRP, clearly showing that enzyme leaching is an important factor affecting the reuse of the membranes produced. This stays in agreement with the data published by Ali et al., who used tyrosinase immobilized by crosslinked polyacrylonitrile (PAN)/chitosan composite membrane for removal of azo dyes. They observed that enzyme elution was the main reason for removal decline, which after 10 cycles reached around 60% for both dyes [30].

Further, storage stability was followed to examine how immobilization affects the long-term activity of the immobilized enzymes. Storage stability of the membrane-immobilized enzymes declined over the storage test, however, all of the membranes retained over 75% of initial relative activity at the end of the storage test, indicating improvement or enzyme stability, as compared to free counterparts. This might be due to the protective effect of the support material as well as enzyme rigidization upon immobilization that protects enzyme structure against conformational changes during storage [2]. Similar observations were made by Li et al., who immobilized laccase onto a nanofiltration membrane made of poly (allylamine hydrochloride) and poly (styrenesulfonic acid) sodium salt. They observed around 90% relative activity retention after 10 days of storage with a future decreasing trend in activity [31].

### 3.2. Removal of Micropollutants from Wastewater

The key aspect of the presented study concerned practical tests of produced biocatalytic membranes in the treatment of wastewater from the domestic treatment plant. Appendix A (data from GC-MS measurements) showed the complexity of the water matrix and the presence of numerous of various compounds of different origins, molecular weight, structure, toxicity, etc. Hence, we decided to follow the removal of six selected compounds commonly used in real-life, industry, and science that frequently occur in analyzed wastewater.

The first stage of the wastewater treatment investigation was the determination of the processing time required to pass the wastewater through the membrane. It was expected that due to the various water permeability of the produced membranes, the process duration for each of the systems tested will be different. In fact, EMR equipped with a membrane-laccase system required around 25% longer process time, as compared to membrane-tyrosinase and membrane-HRP to complete the treatment. This fact is related to higher enzyme loading of this membrane and possible deposition of laccase molecules into membrane pores. This creates additional diffusional limitations, and in consequence, prolonged process time, as reported also by Lante et al. [32]. Although the time required to complete the process seems to be long, taking into account that 20 mL of wastewater was used, this drawback might be overcome by applying higher pressure. It should also be highlighted that process duration depended also on wastewater sampling. A treatment lasting about 10% longer that was observed for wastewater ‘before’ clearly showed that wastewater entering the treatment plant consisted of more substances prone to membrane blocking, even though the preliminary pretreatment was performed. This stays in agreement with previous studies on EMRs in wastewater treatment reported that the more complex the composition of pollutants, the longer process time is required for treatment [33,34,35].

As previously mentioned, the crucial step of this study was the treatment of various wastewater in EMRs equipped with produced systems. All of the treatment experiments using proposed membranes were performed at 4 bars using 20 mL of wastewater solution. Obtained permeates were subjected to GC-MS analysis to determine changes in concentration of selected wastewater components before and after treatment, and final removal efficiency was calculated based on this. It should also be highlighted that in the experiments with the HRP enzyme, in all of the samples, hydrogen peroxide was added, as this compound act as a cosubstrate for HRP. Hydrogen peroxide is known for its oxidation properties. It has been used for many years in wastewater treatment plants for the reduction of biological and chemical oxygen demand in wastewater [36]. It is also known that H_2_O_2_ could affect the properties of the HRP, and influence membrane stability, as well as react with analyzed compounds. However, this factor must be applied in high concentration or together with other chemical/physical factors to significantly influence the analyzed compounds. Literature reports are consistent, stating that additional factors, such as UV radiation, ozone, or micro-nono bubble, are necessary to create a sufficient number of reactive oxygen radicals in the environment to oxidize 2,4-D, EE2, tetracycline, and ketoprofen [37,38,39]. Moreover, porphyrins, such as hematoporphyrin, were proven to produce H_2_O_2_ at high concentrations of porphyrin [40], hence being stable at high hydrogen peroxide concentrations.

When analyzing obtained data, it is obvious that the type of the biocatalytic system used, the type of the micropollutant, as well as wastewater origin, strongly affect the removal rate of the analyzed compounds. About 5–10% higher removal efficiency of all tested pollutants was observed for wastewater ‘after’. This might be explained by the fact that wastewater ‘before’ is characterized by a more complex composition and the presence of compounds that could block enzyme active sites and might lead to enzyme denaturation. A similar observation has been also presented by Nguyen et al., who showed that depending on the composition of the wastewater matrix, the removal rate of the pharmaceutical micropollutants performed in an enzymatic packed-bed reactor varies by about 40% [7]. This observation supports our findings and indicates that proper initial wastewater treatment might significantly improve the catalytic action of the enzymes in the conversion of toxic compounds. From the presented data, it is clear that the type of enzyme deposited within the membrane also affects the final removal efficiency of all compounds. The highest removal rate, irrespectively of the analyzed compounds was observed for membrane with immobilized laccase followed by membrane-tyrosinase and membrane-HRP systems. One of the possible explanations is the fact that the highest enzyme loading and the highest activity retention were observed for membrane with immobilized laccase, suggesting the robustness of this system in the conversion of pharmaceuticals. On the other hand, the laccase enzyme is characterized by the lowest substrate specificity, making this enzyme capable of efficient conversion of a wide range of substrates, also including nonphenolic derivatives [41]. By contrast, HRP is known for the highest substrate specificity, mainly towards phenolic acids and aromatic phenols, which limits the efficient transformation of non-phenolic compounds [42]. These observations are also supported by the previously published data on the removal of phenolic compounds by immobilized enzymes. For instance, Ameri et al. observed over 90% removal of phenol from model water solution using laccase immobilized onto hierarchical NaY zeolite [43], whereas Pantic et al. noticed less than 75% removal of phenol from water solution using HRP immobilized on macroporous glycidyl-based copolymers [44] clearly showing that type of the enzyme and its substrate specificity strongly affect removal rate of pollutants. Finally, during the analysis of the obtained data, it was found that the type and structure of the micropollutant also influence its removal process. Higher removal rates were obtained for compounds consisting of phenolic rings in their structure (2,4-D, tetracycline, EE2, and ketoprofen methyl ester). This was expected as oxidoreductase enzymes applied in this study show a high affinity towards this type of compound [2]. However, the differences in the removal efficiency between these pollutants might be explained by the size and complexity of their molecules and resistance to enzymatic degradation, as well as additional stabilization, provided for instance by the internal transfer of the electrons in the substrate molecule [45]. The expected lower conversion of tert-amyl alcohol and hematoporphyrin could be explained mainly by their structure. Tert-amyl alcohol is characterized by its small size and aliphatic nature which limited its removal. By contrast, hematoporphyrin consists of porphyrin structure in the molecules, which stabilized the structure of this compound and hindered access of these molecules to the active sites of the enzymes. However, to improve the removal rate of these compounds mediators could be applied to improve enzyme-substrate electron transfer and enhance degradation efficiency.

In the presented study, we have also performed control experiments concerning the removal of pollutants by pristine membrane and membrane with thermally inactivated enzymes. Collected data showed that the degradation of analyzed compounds occurred as a synergistic effect of membrane rejection, adsorption, and catalytic conversion, however, with the predominance of enzymatic action. Removal by membrane rejection did not exceed 10% for all tested compounds due to the fact that membrane pores are simply too big to retain toxic molecules. Since the membrane surface is saturated by the enzyme molecules, adsorption onto the membrane surface or into its pores is also limited. The highest value of adsorption was observed for tetracycline, of which adsorption rate by the membrane-laccase system from wastewater “after” reached 18%, whereas for wastewater “before” it was 14%. Generally, due to less complex structure of the wastewater matrix and lower competition for active centers available for adsorption, higher values of adsorption were noticed for wastewater “after”. Moreover, the lowest values of this parameter for membrane-HRP systems could be explained by the biggest size of HRP molecules, which cover the membrane and limit the sorption of other molecules. Briefly summarizing collected data, it is clear that because of to the high activity of the immobilized enzyme and well kinetic parameters noticed for the deposited biomolecules, the highest removal rate observed was due to catalytic transformation.

After a detailed characterization of the treatment process, changes in the membrane’s water permeability were examined, as a drop in membrane permeability is usually observed during the separation of a complex mixture such as wastewater. A significant drop, up to 40%, in water permeability of all tested membranes after treatment of wastewater ‘before’ was noticed, whereas for experiments with wastewater ‘after’, a decline of around 10%was observed, as compared to the membranes after immobilization. This drop is directly related to the complexity of the wastewater matrix, which contains numerous of various compounds that block pores of the membrane and create fouling onto the membrane surface [46]. These corroborate results published by Vitola et al., who used phosphotriesterase-loaded membranes for the removal of pesticides in vegetative waters coming from the olive mill. They observed an over 40% reduction of water permeability mainly due to the presence of solid impurities in the treated solution [47]. It clearly showed that the composition of the wastewater might strongly affect the properties of the membrane by its deposition onto/into the membrane and fouling formation.

## 4. Materials and Methods

### 4.1. Chemicals and Materials

Cellulose filter paper, cellulose membrane at molecular weight cut-off (MWCO) of 20 kDa, and Ultracel membrane at 5 kDa MWCO made of regenerated cellulose were delivered by Merck Millipore Company (Burlington, MA, USA). Peroxidase from horseradish (HRP, EC 1.11.1.7, ≥200 U/mg), tyrosinase (EC 1.14.18.1, ≥10,000 U/mg), laccase (EC 1.10.3.2, ≥0.5 U/mg), hydrogen peroxide (H_2_O_2_, 30%), 2,2′-azino-bis(3-ehylbenzothiazoline-6-sulfonic acid), diammonium salt (ABTS), 50 mM phosphate buffer, and 50 mM acetate buffer at desired pH as well as Bradford reagent were supplied by Sigma-Aldrich (St. Louis, MO, USA). All experiments concerning the use of membranes for immobilization or treatment experiments were performed using Amicon 8050 filtration cell (working area of 13.4 cm^2^) delivered by Sterlitech Company (Auburn, WA, USA) under the pressure of 4 bar unless specified otherwise.

### 4.2. Composition of Sewage from the Wastewater Treatment Plant

In the study, wastewater obtained from a local treatment plant was used. As the composition of wastewater varies depending on the stage of treatment, wastewater was sampled at the entrance to the treatment plant (wastewater ‘before’) and after the treatment process (wastewater ‘after’). The pH of the wastewater ‘before’ was determined to be 4.5, whereas for wastewater ‘after’, the pH was examined as 5.0. Prior to treatment experiments obtained wastewater was filtrated using filter paper, to remove solid impurities and next using a 20 kDa ultrafiltration membrane to remove compounds at a higher molecular weight. Obtained in this way wastewater was stored in dark bottles at ambient temperature until use. The composition of the wastewater was determined using gas chromatography supplemented with a mass spectrometer (GC-MS) and is presented in Appendix A, whereas the estimated initial concentration of the analyzed compounds is presented in Table 3.

### 4.3. Fabrication of Biocatalytic Membranes

The process of fabrication of biocatalytic membranes consisted of enzyme immobilization by passing the enzyme stock solution of the desired biocatalysts through the pristine Ultracell regenerated cellulose membrane, previously moistened with distilled water, with a pore size of 5 kDa, in an Amicon 8050 membrane reactor at the pressure of 2 bar with the agitation of 250 rpm. In each of the immobilization experiments, 20 mL of the 0.5 mg/mL enzyme stock solution was used. Laccase solution was prepared in 50 mM acetate buffer at pH 5, whereas tyrosinase and HRP solutions were performed using 50 mM phosphate buffer at pH 7. The immobilization was finished when the whole volume of the enzyme solution was passed through the membrane. The collected permeates were subjected to further analysis. In this way, the membrane-laccase, membrane-tyrosinase, and membrane-HRP systems were produced and used in further experiments.

### 4.4. Determination of Water Permeability and Flux

The water permeability (*L_p_*) and flux (*J*) were determined for the pristine membrane, membrane after enzyme immobilization, and after the treatment process. For this purpose, the membrane was placed in an Amicon 8050 cell, and the distilled water was used as a medium. The water permeability was calculated according to Equation (1), whereas flux using Equation (2):(1)$$L_{p}=\frac{J_{p}}{TMP}$$
(2)$$J=\frac{V_{p}}{t\cdot A_{m}}$$
where *V_p_* denotes the permeate volume corresponding to experiment time (*t*), *A_m_* denotes the working filtration area, and *TMP* is the transmembrane pressure.

### 4.5. Characterization of Biocatalytic Membranes

The catalytic activity of all of the produced biocatalytic membranes was examined using a model reaction of ABTS oxidation. Briefly, the tested membrane was placed in an Amicon 8050 membrane reactor in darkness (the reactor was covered by aluminum foil) to which 10 mL of ABTS solution at 0.1 mM concentration was added. Experiments with tyrosinase and HRP were performed using 50 mM phosphate buffer at pH 7, whereas reactions with laccase were carried out using 50 mM acetate buffer at pH 5 as a medium. To the system with immobilized HRP 10, mM of H_2_O_2_ was added as a co-substrate for this enzyme. The reaction was stopped after the collection of 8 mL of permeate. Permeate was subjected to spectroscopic measurements at 420 nm (Jasco V-750 spectrophotometer, Tokyo, Japan). To determine the concentration of the ABTS after reaction, a standard ABTS calibration curve was used. In the calculation 1U of enzyme activity was defined as the amount of biocatalyst oxidizing 1 mmol of ABTS per minute under optimal process conditions. Similar tests, concerning the use of the corresponding amount of the free enzyme, were also performed for comparison and calculations. Based on the obtained values of enzymatic activity of free and immobilized enzymes and activity of the permeate after immobilization, activity recovery and immobilization yield were calculated using Equations (3) and (4).
(3)$$\mathrm{Immobilization}\;\mathrm{yield}\;(\%)=\frac{A_{i}-A_{p}}{A_{i}}\cdot 100\%$$
(4)$$\mathrm{Activity}\;\mathrm{recovery}\;(\%)=\frac{A_{t}}{A_{i}}$$
where *A_i_* denotes the initial activity of the enzyme in the solution used for immobilization, *A_p_* denotes the activity of the enzyme in the permeate after immobilization and *A_t_* denotes the activity of the immobilized enzyme.

The amount of the immobilized enzyme and consequently membrane loading was calculated according to the Bradford method [48] based on the spectrophotometric measurements at 595 nm (Jasco V-750 spectrophotometer, Japan). For the measurement, 0.1 mL of the Bradford reagent was mixed with 0.1 mL of deionized water and 0.8 mL of the analyzed solution (enzyme stock solution and permeate after immobilization). After 10 min, the UV-Vis measurements were made. The amount of immobilized enzyme (mg) is considered as the difference between the initial and the final amount of the enzyme in the solution before and after immobilization, respectively. The membrane loading (mg/cm^2^) was calculated by dividing the amount of immobilized enzyme by the working area of the membrane.

Storage stability of the produced membranes was examined based on the above-mentioned method using ABTS as a substrate at optimal process conditions for each of the membrane. The relative activity of free and immobilized enzyme was determined every specified period of time over 20 days of membranes storage at 4 °C. The initial activity of free and immobilized enzymes was set as 100% relative activity.

The reusability of the produced membranes was determined over 10 repeated model reaction cycles with 10 mL of ABST as a substrate, at conditions optimal for each of the membrane. After each reaction, cycle membrane was washed twice with distilled water and fresh ABTS solution was transferred into reactor. During reusability study, enzyme leaching from the membrane was determined as well. For this reason, permeate collected after reaction step and washing solutions collected after membrane rising were subjected into UV-Vis measurements according to Bradford method. Enzyme loading (%) is considered the mass of the enzyme that is retained within the membrane after each repeated use.

The kinetic parameters of the free and immobilized enzyme (Michaelis–Menten constant (*K_M_*) and the maximum reaction rate (*V_max_*) were determined based on the model ABTS oxidation reaction with substrate solutions range adjusted for each of the enzyme. For examination of kinetic parameters, produced membranes were cut into 0.5 × 0.5 cm pieces and tested into smaller scale under optimal process conditions. The Hanes–Woolf plot was applied to calculate the apparent kinetic parameters of free and immobilized enzyme.

### 4.6. Enzymatic Treatment of Wastewater in EMR

To perform enzymatic treatment of wastewater, produced biocatalytic membranes were placed in an Amicon 8050 filtration cell, to which 20 mL of the various wastewater solution was added. To each of the wastewater samples, 1 mL of 5 mM ABTS solution was added as a mediator to enhance enzyme activity. Experiments were carried out at 25 °C, 4 bars with mixing at 250 rpm. The process was completed when the total volume of the wastewater was passed through the membrane. Hence, the duration of single experiments depends on type of the wastewater and biocatalytic membrane applied. After whole permeate volume was collected, it was subjected into GC-MS analysis to examine removal efficiency (*RE*, %) of the selected compounds, including hematoporphyrin (porphyrin), (2,4-dichlorophenoxy)acetic acid (2,4-D, herbicide), 17α-ethynylestradiol (EE2, estrogen), tetracycline (antibiotic), tert-amyl alcohol (anesthetic drug) and ketoprofen methyl ester (nonsteroidal anti-inflammatory drug), and this was calculated according to Equation (5). These compounds were selected to detail analysis, as they were detected in both, wastewater “before” and “after”, and they represent various groups of compounds. Further, all of these compounds presented in uncontrolled amount in wastewater are generally considered micropollutants.
(5)$$\mathrm{RE}\;(\%)=\frac{C_{I}-C_{F}}{C_{I}}$$
where *C_I_* and *CF* denote the concentration of the selected compound before and after removal, respectively.

In order to perform control experiments for the removal of pollutants and to determine if the removal of pollutants is a simultaneous action of membrane rejection membrane adsorption and catalytic conversion, a series of experiments were performed. For this reason, three systems were examined and compared: (i) pristine membrane (membrane rejection), (ii) membrane with enzyme subjected to thermal inactivation (4 h at 80 °C; membrane rejection and adsorption of pollutants), and (iii) fully active biocatalytic membrane (simultaneous removal via enzymatic conversion, adsorption, and membrane rejection). The removal experiments for all of the biocatalytic systems produced and for all tested compounds were performed according to the methodology presented above in this section under optimal process conditions (25 °C, 4 bars, 250 rpm mixing) using wastewater “before” and “after”. The final removal rate of pollutants by proposed membranes was examined separately for each system using GC-MS analysis followed by the determination and comparison of the percentage contribution of each technique.

### 4.7. Analytical Techniques

Bruker Vertex 70 spectrometer (Bruker, Bremen, Germany) working in attenuated total reflectance (ATR) mode with diamond crystal was applied to obtain Fourier transform infrared spectra (FTIR) of pristine membrane, free enzyme, and membranes after enzyme immobilization. The measurements were conducted in the wavenumber range of 4000 to 400 cm^−1^, at a resolution of 0.5 cm^−1^ and with 64 scans.

The preparation of samples for GCMS analysis consisted in the first step of extracting 1 mL of a 1:1 mixture of ether and hexane (High Purity Grade, Merck, Darmstadt, Germany). For this, the described mixture was added to each of the samples, followed by vortexing for 90 s. After the emulsion separation, the upper phase was collected and then subjected to GCMS analysis. The GC–MS system used for analysis was a Pegasus 4D GCxGC-TOF MS from LECO (Leco Corp., Benton Harbor, MI, USA). Data acquisition and analysis were performed using standard software supplied by the manufacturer. Substances were separated on a BPX5 capillary column (60 m × 0.25 mm ID, 0.25 µm film thickness) (Trajan Scientific and Medical, Ringwood, Australia). Temperature program: 35 °C hold for 2 min, 7 °C/min up to 300 °C, hold for 8 min. The temperatures for the injection port and transfer line were set both at 250 °C. Split injection mode (1:49) and helium with a flow rate of 1 mL/min as carrier gas were used.

### 4.8. Statistical Analysis

Membrane experiments were performed in triplicate. Results are presented as means ± standard deviations. One-way ANOVA performed in SigmaPlot 12 (Systat Software Inc., San Jose, CA, USA) using Tukey’s test was applied to examine statistically significant. Statistical significance was established at *p* < 0.05.

## 5. Conclusions

In the presented study we proposed production and application of the membranes with immobilized oxidoreductases for wastewater treatment. Efficient production of above-mentioned systems was confirmed by the results of FTIR analysis. Thorough analysis of the produced materials, it was made evident that laccases immobilized within membrane showed the highest enzyme loading and activity recovery. Produced membranes that were exhibited also improved long-term stability as well as reusability, as compared to free enzymes. It was also found that immobilized enzymes showed lower substrate affinity and are characterized by lower water permeability and flux, as compared to the pristine cellulose membrane. However, practical tests in the wastewater treatment showed that proposed systems were found effective in the removal of selected pollutants from wastewater. Over 50% of 2,4-D and tetracycline was removed by membrane-laccase system, followed by over 40% removal of EE2 and ketoprofen methyl ester and over 20% degradation of tert-amyl alcohol hematoporphyrin. Collected data clearly showed that various factors affected wastewater treatment, including type of the used enzyme and sapling place of wastewater, as well as type of compound to be degraded. Although collected data indicated produced systems as efficient, further studies are still highly required in order to transfer presented approach into larger scale and to improve removal of compounds resistant to enzymatic conversion.

## Figures and Tables

**Figure 1 ijms-23-14086-f001:**
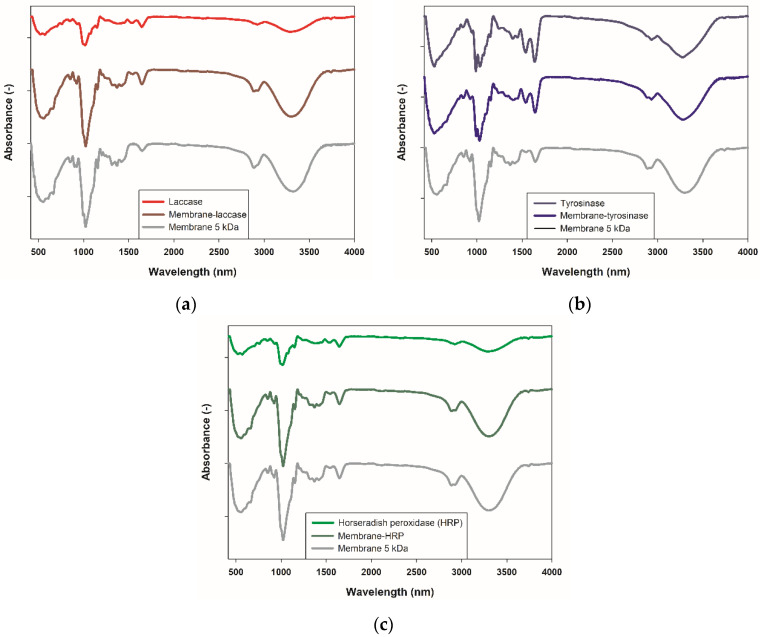
Fourier transform infrared spectra of the pristine cellulose membrane and (**a**) free laccase and membrane after laccase immobilization, (**b**) free tyrosinase and membrane after tyrosinase immobilization and (**c**) free horseradish peroxidase (HRP) and membrane after HRP immobilization.

**Figure 2 ijms-23-14086-f002:**
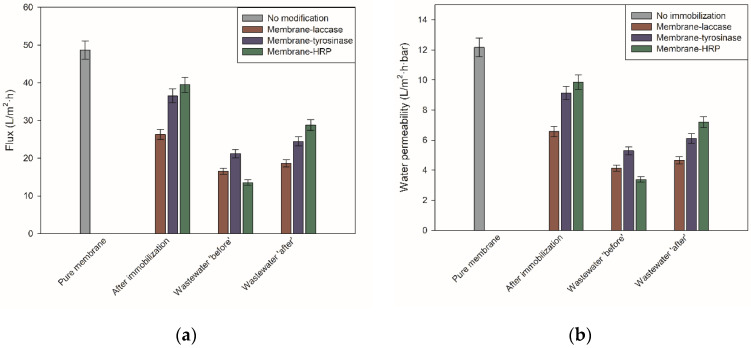
Characterization of the biocatalytic membranes produced in terms of changes in their (**a**) flux and (**b**) water permeability. All values are presented as a mean value of 3 experiments ± standard deviation.

**Figure 3 ijms-23-14086-f003:**
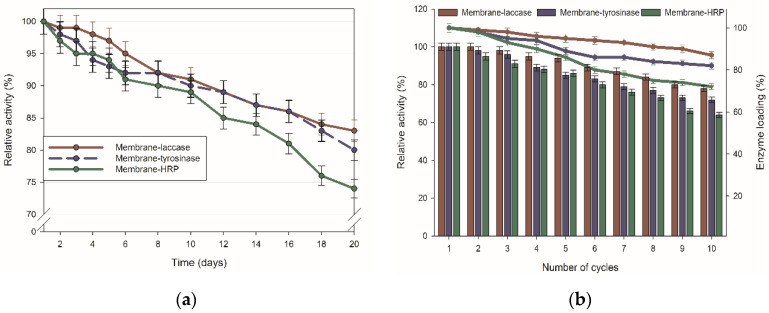
(**a**) Storage stability as well as (**b**) reusability (bars) and enzyme loading (points) over repeated use of the produced biocatalytic membranes. All values are presented with a mean value of 3 experiments ± standard deviation.

**Figure 4 ijms-23-14086-f004:**
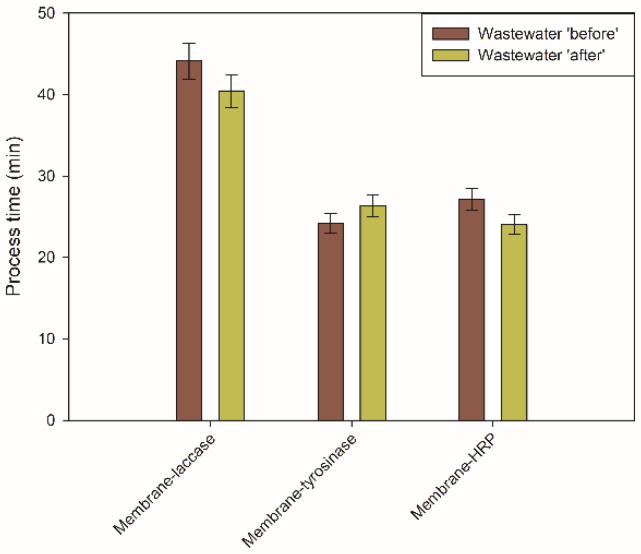
Process duration of the wastewater treatment experiments by various biocatalytic systems produced. All values are presented with a mean value of 3 experiments ± standard deviation.

**Figure 5 ijms-23-14086-f005:**
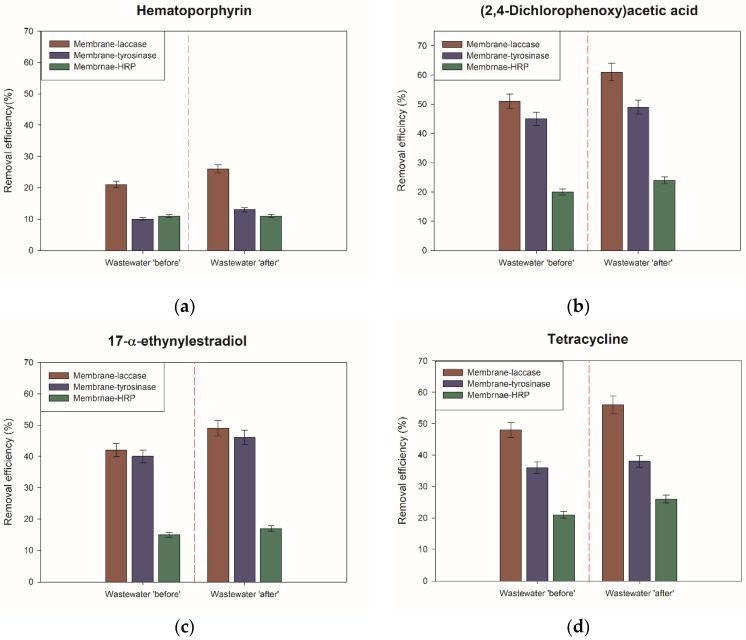
Removal efficiency of (**a**) hematoporphyrin, (**b**) (2,4-dichlorophenoxy)acetic acid (2,4-D), (**c**) 17α-ethynylestradiol (EE2), (**d**) tetracycline, (**e**) tert-amyl alcohol and (**f**) ketoprofen methyl ester after treatment by various biocatalytic membranes. All values are presented with a mean value of 3 experiments ± standard deviation.

**Table 1 ijms-23-14086-t001:** Characterization of enzyme immobilization including calculation of immobilization yield, amount of immobilized enzyme, and membrane loading. All values are presented as a mean value, and which error value does not exceed 5%.

Analyzed Parameter	Membrane-Laccase	Membrane-Tyrosinase	Membrane-HRP
Immobilization yield (%)	91	82	85
Amount of immobilized enzyme (mg)	9.1	8.2	8.5
Membrane loading (mg/cm^2^)	0.68	0.61	0.63

**Table 2 ijms-23-14086-t002:** Characterization of biocatalytic membranes produced in terms of activity recovery and kinetic parameters of free and immobilized enzymes. All values are presented as mean values, and which error value does not exceed 5%.

Analyzed Parameter	Membrane-Laccase	Membrane-Tyrosinase	Membrane-HRP
Activity recovery (%)	90%	85%	82%
Michaelis-Menten constant (*K_M_*, mM)	0.094	0.81	1.52
Maximum reaction rate (*V_max_*, mM/min)	0.103	104	298

**Table 3 ijms-23-14086-t003:** Estimated concentration of various substances in wastewater ‘before’ and ‘after’ obtained from local wastewater treatment plant subjected to detailed analysis.

Analyzed Compound	Group	Wastewater ‘Before’	Wastewater ‘After’
		Estimated Concentration Range (ng/mL)	
Hematoporphyrin	porphyrin	0.1–1	0.1–1
(2,4-Dichlorophenoxy) acetic acid	herbicide	1–10	0.1–1
17α-Ethynylestradiol	estrogen	0.1–1	0.1–1
Tetracycline	antibiotic	0.1–1	0.1–1
tert-Amyl alcohol	anesthetic drug	0.1–1	0.1–1
Ketoprofen methyl ester	nonsteroidal anti-inflammatory drug’s ingredient	1–10	0.1–1

## Data Availability

The data presented in this study are available on request from the corresponding author.

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
