# Peer review of "Study of Membrane-Immobilized Oxidoreductases in Wastewater Treatment for Micropollutants Removal"

_ijms, 2022, doi:10.3390/ijms232214086_

Round 1

Reviewer 1 Report

General:

In the paper “Study of Membrane-immobilized Oxidoreductases in Wastewater Treatment for Micropollutants Removal”, the authors examined the feasibility of three enzyme immobilization on commercially available cellulose membranes. The activity of regenerated membrane and after long-term storage (up to 20 day) is characterized. The paper is very well written and the results are solid. One major concern and several minor comments are listed below.

Major:

As for three oxidoreductases, are there any mediators being added to the system? 

Minor:

(1)    Figure 3, in the main text, it would be better to briefly mention that the membrane-enzyme activity was determined using a model reaction of ABTS oxidation (as mentioned in the methods part), so the reader can have the first impression of how this activity is characterized.

(2)    Figure 3B, same as in comment 1, in the main text, it would be very helpful to mention how the loss of the enzyme on the membrane was determined (i.e., Bradford method)?

(3)    For immobilized HRP, H2O2 was also added as a co-substrate for this enzyme. Will H2O2 also react with the compounds/MPs? It would be better to add the discussion into the main text.  

Author Response

Reviewer 1

In the paper “Study of Membrane-immobilized Oxidoreductases in Wastewater Treatment for Micropollutants Removal”, the authors examined the feasibility of three enzyme immobilization on commercially available cellulose membranes. The activity of regenerated membrane and after long-term storage (up to 20 day) is characterized. The paper is very well written and the results are solid. One major concern and several minor comments are listed below.

Query 1: As for three oxidoreductases, are there any mediators being added to the system?

Answer 1: Yes, to each of the wastewater samples subjected to remediation, an ABTS was added as a mediator. The following statement was provided in Section 4.6 to clearly present data on ABTS: “To each of the wastewater samples, 1 mL of 5 mM ABTS solution was added as a mediator to enhance enzyme activity.”

Query 2: Figure 3, in the main text, it would be better to briefly mention that the membrane-enzyme activity was determined using a model reaction of ABTS oxidation (as mentioned in the methods part), so the reader can have the first impression of how this activity is characterized.

Answer 2: Thank the Referee for this suggestion. We have added the following information in the text to better introduce readers to the issue: “In the study, the activity of the enzymes
immobilized using membranes over storage time and repeated use was determined using a model oxidation reaction of ABTS.”

Query 3: Figure 3B, same as in comment 1, in the main text, it would be very helpful to mention how the loss of the enzyme on the membrane was determined (i.e., Bradford method)?

Answer 3: In the body of the manuscript following text has been added in Section 2.2. to introduce the reader to the issue: “In the study, amount of enzyme leaked from the support was determined according to Bradford methodology by spectroscopic analysis of permeates after each degradation cycle.” However, a detailed methodology for the determination of enzyme leaching from the membrane is presented in Section 4.5 in the Materials and Methods section.

Query 4: For immobilized HRP, H2O2 was also added as a co-substrate for this enzyme. Will H2O2 also react with the compounds/MPs? It would be better to add the discussion into the main text.

Answer 4: We would like to thank you for this comment. The following text has been added in the revised manuscript to explain how H2O2 could react with the compounds/MPs: “It should also be highlighted that in the experiments with HRP enzyme, in all of the samples, hydrogen peroxide was added, as this compounds act as a cosubstrate for HRP. Hydrogen peroxide is known for its oxidation properties. It has been used for many years in wastewater treatment plants for the reduction of biological and chemical oxygen demand in wastewater [36]. It is also known that H2O2 might affect the properties of the HRP, could influence membrane stability as well as might react with analyzed compounds. However, this factor must be applied in high concentration or together with other chemical/physical factors to significantly influence the analyzed compounds. Literature reports are consistent, that additional factors such as UV radiation, ozone, or micro-nono bubble are necessary to create a sufficient number of reactive oxygen radicals in the environment to oxidize 2,4-D, EE2, tetracycline, and ketoprofen [37-39]. Moreover, porphyrins, such as hematoporphyrin were proved to produce H2O2 at high concentrations of porphyrin [40], hence being stable at high hydrogen peroxide concentration.”

Reviewer 2 Report

The manuscript ijms-1942594 describes the removal of a number of emerging micropollutants by the action of oxidoreductase enzymes immobilized on cellulosic membrane material. The approach of enzyme immobilization utilized in this study are indeed novel and therefore, the findings of this work is significant. It will also be of interest to the readership of IJMS. However, I see several major pitfalls within the study design and the conclusions drawn from the observations. If these can be addressed, this work will stand out as an important piece of work within the field of enzyme immobilization for biodegradation of micropollutants. Please consider the following;

1. The major problem in this study is the absence of control experiments. Nowhere in the manuscript mentions the inclusion of control experiments during the study design and analysis. A proposed control experiment is the use of cellulosic membrane matrix on its own (without and immobilized enzymes) and inclusion of the micropollutants at the same concentrations as used in other tests. The same set of quantitative analyses to determine the before and after concentrations of the micropollutants should also be performed. 

2. An adsorption study for the micropollutants should also be performed on the control experiments. We do not know the what proportion of the micropollutant removal is taking place due to simple adsorption on the cellulosic membranes. Please model adsorption patterns using a mathematical model such as a Langmuir adsorption isotherm and report the appropriate parameters in the results and discussion.

Author Response

Reviewer 2

The manuscript ijms-1942594 describes the removal of a number of emerging micropollutants by the action of oxidoreductase enzymes immobilized on cellulosic membrane material. The approach of enzyme immobilization utilized in this study are indeed novel and therefore, the findings of this work is significant. It will also be of interest to the readership of IJMS. However, I see several major pitfalls within the study design and the conclusions drawn from the observations. If these can be addressed, this work will stand out as an important piece of work within the field of enzyme immobilization for biodegradation of micropollutants. Please consider the following;

Query 1: The major problem in this study is the absence of control experiments. Nowhere in the manuscript mentions the inclusion of control experiments during the study design and analysis. A proposed control experiment is a use of a cellulosic membrane matrix on its own (without and immobilized enzymes) and the inclusion of the micropollutants at the same concentrations as used in other tests. The same set of quantitative analyses to determine the before and after concentrations of the micropollutants should also be performed.

Query 2: An adsorption study for the micropollutants should also be performed on the control experiments. We do not know what proportion of the micropollutant removal is taking place due to simple adsorption on the cellulosic membranes. Please model adsorption patterns
using a mathematical model such as a Langmuir adsorption isotherm and report the appropriate parameters in the results and discussion.

Answer 1 and 2: We would like to thank the Referee for these valuable comments. According to the suggestion, we have performed additional control experiments with the same concentration of analyzed compounds, concerning the removal of pollutants by a pristine membrane (rejection of pollutants by the membrane) and membrane with thermally inactivated enzyme (adsorption of the pollutants). The following text has been introduced in the Materials and Methods section to describe performed experiments: “In order to perform control experiments for the removal of pollutants and to determine if removal of pollutants is a simultaneous action of membrane rejection membrane adsorption and catalytic conversion, a series of experiments were performed. For this reason, three systems were examined and compared: (i) pristine membrane (membrane rejection), (ii) membrane with enzyme subjected to thermal inactivation (4 h at 80 °C; mem-brane rejection and adsorption of pollutants), and (iii) fully active biocatalytic membrane (simultaneous removal via enzymatic conversion, adsorption, and membrane rejection). The removal experiments for all of the biocatalytic systems produced and for all tested compounds were performed according to the methodology presented above in this section under optimal process conditions (25 °C, 4 bars, 250 rpm mixing) using wastewater “before” and “after”. The final removal rate of pollutants by proposed membranes was examined separately for each system using GC-MS analysis followed by determination and comparison of the percentage contribution of each technique.” From the obtained data (see Supplementary materials, Table S3 I Table S4) it is clear that the removal of pollutants occurred due to the simultaneous action of membrane rejection, adsorption, and catalytic conversion, however, enzymatic action should be considered as the dominant process in the degradation process of all compounds. In the Results section, we have provided the following text to describe data obtained from control tests: “In the study also control experiments with pristine membrane and membrane with thermally inactivated enzyme have been performed to determine the contribution of every single process in total removal efficiency. Collected data (Supplementary materials Table S3 and Table S4) clearly showed that rejection of pollutants by membrane ranged from 3% for hematoporphyrin to 10% for 2,4-D. Slightly higher results of pollutant removal were observed for membranes with the thermally inactivated enzyme. The highest removal by adsorption (17%) was observed for 2,4-D for the membrane-tyrosinase system, whereas the lowest (2%) for tert-amyl alcohol adsorbed on membrane-laccase and mem-brane-HRP system. Further, higher removal rates, of around 3% to 5% of all tested pollutants were noticed from wastewater “after” treatment. Finally, from the obtained data it is clear that catalytic conversion was the main driving mechanism of pollutants removal. The removal rate by membranes with active enzymes usually exceeded 20%, reaching up to 40% for the membrane-laccase system. By contrast, the lowest percentage contribution of catalytic removal was observed for membranes with immobilized HRP. In this case, removal of hazardous compounds did not exceed 10%.” We have also added proper text in the Discussion section to explain why enzymatic conversion is
the main mechanism of degradation: “In the presented study we have also performed control experiments concerning the removal of pollutants by pristine membrane and membrane with thermally inactivated enzymes. Collected data showed that degradation of analyzed compounds occurred as a synergistic effect of membrane rejection, adsorption, and catalytic conversion, however, with a predominance of enzymatic action. Removal by membrane rejection did not exceed 10% for all tested compounds due to the fact that membrane pores are simply too big to retain toxic molecules. Since the membrane surface is saturated by the enzyme molecules, adsorption onto the membrane surface or into its pores is also limited. The highest value of adsorption was observed for tetracycline, which adsorption rate by the membrane-laccase system from wastewater “after” reached 18%, whereas for wastewater “before” it was 14%. Generally, due to the less complex structure of the wastewater matrix and lower competition for active centers available for adsorption, higher values of adsorption were noticed for wastewater “after”. Moreover, the lowest values of this parameter for mem-brane-HRP systems could be explained by the biggest size of HRP molecules, which covers the membrane and limit the sorption of other molecules. Briefly summarizing collected data, it is clear that due to high activity of the immobilized enzyme and well kinetic parameters noticed for the deposited biomolecules, the highest removal rate was observed due to catalytic transformation.” Finally, we would like to add that due to the very limited percentage contribution of the adsorption process in the total removal rate of micropollutants and because the presented manuscript is focused on enzymatic conversion, we decided not to deeply analyze the adsorption process itself. However, we are grateful for this comment, as in our next manuscript we will more focus on the detailed analysis of the adsorption process.
